# Cerebral Resistance Artery Histological Remodeling After Training—Sex Differences

**DOI:** 10.3390/life15081304

**Published:** 2025-08-17

**Authors:** Tobias Hainzl, György L. Nádasy, Emese Róza Márka, Kamilla Nagy, Réka Kollarics, Anna-Mária Tőkés, Attila Oláh, Tamás Radovits, Béla Merkely, Nándor Ács, Szabolcs Várbíró, Attila Jósvai, Marianna Török

**Affiliations:** 1Doctoral School, Semmelweis University, 1091 Budapest, Hungary; hainzl.tobias@semmelweis.hu; 2Department of Internal Medicine and Oncology, Semmelweis University, 1083 Budapest, Hungary; 3Department of Physiology, Semmelweis University, 1094 Budapest, Hungary; nadasy.gyorgy@med.semmelweis-univ.hu; 4Department of Obstetrics and Gynaecology, Semmelweis University, 1082 Budapest, Hungary; marka.emese@stud.semmelweis.hu (E.R.M.); nagy.kamilla99@stud.semmelweis.hu (K.N.); acs.nandor@semmelweis.hu (N.Á.); varbiro.szabolcs@semmelweis.hu (S.V.); 5Department of Obstetrics and Gynaecology, University of Szeged, 6725 Szeged, Hungary; kollarics.reka@phd.semmelweis.hu; 6Department of Pathology, Forensic and Insurance Medicine, Semmelweis University, 1091 Budapest, Hungary; tokes.anna.maria@semmelweis.hu; 7Heart and Vascular Center, Department of Experimental Cardiology and Surgical Techniques, Semmelweis University, 1122 Budapest, Hungary; olah.attila@semmelweis.hu (A.O.); radovitstamas@yahoo.com (T.R.); 8Heart and Vascular Center, Department Cardiology, Semmelweis University, 1122 Budapest, Hungary; merkely.bela@gmail.com; 9Workgroup for Science Management, Doctoral School, Semmelweis University, 1085 Budapest, Hungary; 10Department of Neurosurgery, Military Hospital, Hungary, 1134 Budapest, Hungary; dr.josvai.attila@gmail.com

**Keywords:** cerebral vessels, morphological changes, exercise training, sex differences, colorimetry

## Abstract

**Background**: Chronic exercise has been linked to positive effects on cognitive function and brain health. The aim of our study was to investigate how exercise affects cerebral resistance artery morphology, with an underlying focus on potential sex differences. **Methods**: Wistar rats were divided into male exercising (M.Ex; *n* = 6), female exercising (F.Ex; *n* = 5), male sedentary (M.Sed; *n* = 5), and female sedentary (F.Sed; *n* = 5) groups. After a 12-week swimming program, histological examinations of the intracerebral and pial arterioles were performed. SMA-DAB (smooth muscle actin) and resorcin-fuchsin (elastica) stained brain coronal sections were used for quantitative colorimetric analysis. **Results**: Investigating the effect of exercise, we found that in both pial and intracerebral arterioles, the elastic fiber density increased in both female and male exercising animals compared to the sedentary groups (*p* < 0.05 (M.Sed vs. M.Ex); *p* < 0.0001 (F.Sed vs. F.Ex)). As sex differences, we found that in female animals’ pial arterioles, the density of elastic fiber was increased compared to the male exercising group (*p* < 0.001 (M.Ex vs. F.Ex)). In pial arterioles, the smooth muscle density was higher in the male sedentary animals (*p* < 0.01 (M.Sed vs. F.Sed)); in intracerebral arterioles, the smooth muscle density increased with exercise in the male animals as well (*p* < 0.0001 (M.Ex vs. F.Ex)). **Conclusions**: Our results demonstrate that the increase in vascular elasticity is more pronounced overall in female animals.

## 1. Introduction

Regular exercise benefits cardiovascular health, reducing the risk of major cardiovascular events in healthy individuals and lowering disease incidence in at-risk patients. Its effects include lower body weight, reduced blood lipids, increased muscle mass, increased glucose metabolism and insulin sensitivity, as well as enhanced endothelial dilation and improved hemodynamics [1].

Cerebral blood flow remains relatively stable during physical activity despite increases in cardiac output and systemic blood pressure due to blood vessel regulatory mechanisms in both macro- and microcirculation, including myogenic, metabolic, and neurogenic regulation [2]. The impact of exercise on blood flow, structure, and function in cerebral vessels has been widely studied [3,4], especially in the context of aging and disease.

Research shows that regular physical activity improves cognitive function across all ages, supports stroke recovery, and may slow dementia progression [5,6,7,8,9,10]. Aging leads to arterial stiffening and altered blood flow, which intensive exercise may prevent or even reverse [11,12,13,14,15,16]. Sex differences exist in cognitive decline and exercise effects, with women—especially premenopausal—being more protected against cardiovascular events [17,18,19]. Previous studies targeted on sex differences in vascular function mainly focused on peripheral blood vessels and less on cerebral blood vessels [20,21,22,23], with limited research on cerebral blood vessel reactivity to exercise [20,24]. While the positive functional effects of chronic exercise are well-documented [25,26,27], their histological basis remains mostly unexplored.

Based on the above, our hypothesis was that exercise would result in histological changes in the wall structure of intracerebral and pial vessels compared to sedentary animals, as our group observed in previous studies of other organs [28,29,30]. In addition, the question was raised whether we could also identify sex differences. The relevance of this issue is shown by the fact that the Scientific Statement of the American Heart Association of 2024 has a specific paragraph dedicated to the beneficial effects of exercise in women [27]. The beneficial effects of exercise on cognitive function, cerebral circulation, and cerebral circulation regeneration have also been investigated in several studies [25,26,27]. However, the histological changes associated with functional changes have not yet been described. Our study aims to fill this gap.

## 2. Materials and Methods

### 2.1. Animals

12-week-old young adult male (*n* = 11) and female (*n* = 10) Wistar rats were kept in conventional cages at a constant temperature (22 + 2 °C) and 12 h day–night rhythm. The animals had free access to standard rat food and water. Animal housing and experimental conditions followed the guidelines of the ‘Guide for the Care and Use of Laboratory Animals’ by the National Institutes of Health (NIH Publication No. 86-23, revised 1996) and the European Union (Directive No. 2010/63/EU). The experiment was approved by the Animal Care Committee of Semmelweis University and Hungarian authorities (permission number: PEI/001/2374-4/2015; approval date: 30 July 2015).

### 2.2. Chemicals

For anesthesia, 45 mg/kg i.p. pentobarbital (Euthasol, CEVA Santé Animale, Liboume, France) was used.

### 2.3. Intensive Swim Training Protocol

After one week of acclimatization, animals were randomly divided into four groups: male exercising (MEx, *n* = 6), female exercising (FEx, *n* = 5), male sedentary (MSe, *n* = 5), and female sedentary (FSe, *n* = 5). A graded, intensive swimming training program was applied to the swimming groups (MEx and FEx) [31]. Rats are good swimmers; water is a natural environment for them. Animals were individually placed in a flat-walled water tank containing relatively warm (30–32 °C) water, which was divided into six lanes, each 20 × 25 cm long and 45 cm deep. The size of the lanes was determined so that the rats could not hit the wall and lean against it while swimming in their lane. The program started with 15 min of swimming each day, and swimming time was increased by 15 min daily until it reached 200 min. That duration was then maintained throughout the experiment [31]. For a total of twelve weeks, the trained rats swam five days a week and rested the other two. The 12-week training program was carried out in parallel with the sedentary groups, MSe and FSe, who swam five minutes a day only, five days a week [31]. No animals were lost nor were any complications encountered during the training program. All animals were healthy throughout the experimental period.

### 2.4. Histology and Immunohistochemistry

Formalin-fixed tissue samples were embedded in paraffin and cut into 5 µm sections. After deparaffinization and endogenous peroxidase blocking, antigen retrieval (Tris-Edta, pH 9) was performed for 30 min. To visualize smooth muscle fibers, a smooth muscle actin (SMA) antibody (Cell Marque) was used in a 1:100 concentration followed by DAB chromogen. Smooth muscle fibers were stained brown in the sections. To detect elastic fibers, a resorcin-fuchsin (RF) staining was used, which stained the elastic fibers a magenta color. Selected coronal brain sections were digitized and studied with the aid of the Case Viewer 2.4. software program (Case Viewer, 3DHISTECH Ltd., Budapest, Hungary).

### 2.5. Image Analysis

A total of 15–30 intracerebral arteriolar cross sections, and the same number of pial arteriolar cross sections, were identified on each stained coronal section at 40× magnification (pixel size 0.25 × 0.25 µm) and photographed. Anything not belonging to the vessel was cut out of the image. A total of 845 SMA-DAB immunostained and 719 RF stained (elastica) vascular cross sections were studied in the four groups. Vascular cross sections were split into different diameter ranges (<15 µm, 15–40 µm, and >40 µm) and subjected to colorimetric analysis. Diameter ranges were based on Sweeny et al.’s classification of cerebral arterioles [32].

Analyses for quantitative colorimetry were run under Python 3.10.4 with packages NumPy 1.23., Pillow 9.4.0, and OpenCV 4.7.0. The diameter of each vessel was calculated based on the image’s pixel dimensions and total area. A scaling factor of 1.25 (20%) was applied to the diameter, accounting for processing-induced shrinkage [33].

SMA-DAB sections were color split: the red values, blue values, and red per blue ratio values were determined for each pixel, and a histogram for red per blue values was constructed. According to previous studies [34], the calibration was made as follows: unanimously elastic and DAB-positive territories, respectively, were cut from stochastically chosen sections (several millions of pixels). For comparison, unanimously non-elastic (or DAB-negative) territories have also been cut from the pictures. Red, green, and blue intensities on the 1–255 color intensity scale have been red for each pixel, and two-dimensional color intensity histograms were constructed. These diagrams demonstrated that for (our standard staining procedure) SMA DAB R/B&gt, 1.2 pixels give a good, objective definition for DAB positivity. In the case of RF, the purple color depresses the green component; G&lt 40 can be defined as containing mostly elastin component. DAB-positive pixels are characterized by red per blue values over 1.25 for weak staining and 1.65 in cases of dense staining. Here, DAB positivity was defined as the ratio being over 1.65. For each vessel, the percentage of the area with dense DAB-positive staining was calculated. For the resorcin-fuchsin (elastica) staining, after color splitting, green pixel histograms were constructed. The magenta color of dense elastica suppresses the green color component at green levels below 40 (BMP RGB 256 color intensity levels). RF-positive staining was defined as below 40, and the percentage of the area with RF-positive staining was calculated for each vessel [34,35]. The RF-positive and SMA-positive percentages were compared across different groups and diameters. The selection of the arterial cross sections and the colorimetric analysis was double blinded.

### 2.6. Statistical Analysis

Statistical analysis and graphing were performed in R.3.4.1 (R Foundation for Statistical Computing, Vienna, Austria), using tidyverse 1.3.1, car 3.0-14, and emmeans 1.8.3 packages. Normality was assessed with the Shapiro–Wilk test (α = 0.05). For datasets meeting normality, a two-way ANOVA with Tukey’s post hoc was performed. For datasets violating the normality assumption, Kruskal–Wallis tests, followed by Dunn post hoc comparisons, were conducted. The criteria for statistical significance were set at *p* < 0.05. Post hoc power calculations were performed for each size category using G*Power 3.1.9.7.

## 3. Results

### 3.1. Exercise Effect of Pial Arterioles

#### 3.1.1. RF Staining

Examining the effect of exercise training on the pial vessels, we found that exercise training increased elastic fiber density in both female and male animals in vessels with a diameter above 15 µm compared to sedentary groups (Figure 1A, statistical power: f > 0.56, power > 89%).

#### 3.1.2. SMA Staining

There was no significant increase in smooth muscle density in the pial arterioles after exercise in either sex compared to sedentary groups (Figure 1B, statistical power: in the <15 µm (f = 0.33, power 28%) and >40 µm (f = 0.55, power 72%) diameter ranges).

### 3.2. Sex Differences in Pial Arterioles

#### 3.2.1. RF Staining

Comparing male and female sedentary groups, no significant difference in elastic fiber density was found between the two groups. Examining the male and female exercising groups, we observed a significant increase in elastic fiber density in female animals compared to the exercising males in diameter ranges above 15 µm (Figure 1A, statistical power: f > 0.56, power > 89%).

#### 3.2.2. SMA Staining

Comparing male and female sedentary animals, we observed that male sedentary animals had a significantly higher smooth muscle density in the 15–40 µm vessel diameter range compared to the female sedentary group. This difference disappears after training. (Figure 1B, statistical power: f = 0.21, power 85%).

### 3.3. Exercise Effect of Intracerebral Arterioles

#### 3.3.1. RF Staining

Training significantly increased the density of elastic fibers in intracerebral arterioles in the 15–40 µm diameter range in both sexes compared to sedentary groups (Figure 2A, statistical power: f = 0.43, power 80%).

#### 3.3.2. SMA Staining

No significant difference in the smooth muscle density of the intracerebral arterioles was found between sedentary and exercising groups in either sex (Figure 2B, statistical power-: in the <15 µm (f = 0.13, 13%) and >40 µm (f = 0.16, power 8%).

### 3.4. Sex Differences in Intracerebral Arterioles

#### 3.4.1. RF Staining

Comparing the male and female sedentary groups, no significant difference in elastic fiber density was found between the two groups. In contrast to the pial vessels, there was no significant difference in elastic fiber density between the male and female exercising groups (Figure 2A).

#### 3.4.2. SMA Staining

Smooth muscle density was significantly higher in male exercising animals compared to exercising female animals in the 15–40 µm diameter range (Figure 2B, statistical power: f = 0.22, power 98%).

## 4. Discussion

In the present study, we investigated the effect of exercise on the histological composition of intracerebral and pial resistance artery vessels, with a focus on sex differences, a question raised by several earlier publications [17,18,19,20,21,22,23,24,36]. Our observations demonstrated significant differences between intracerebral and pial vessels both in sedentary and exercising groups in smooth muscle contractile protein and elastin content. Chronic exercise led to substantial changes in the active and passive force-bearing histological elements in both vessel types. Sexual differences in vessels were present in the sedentary state, and even the exercise-induced alterations were different in males and females.

Depending on the intensity of exercise, systemic systolic blood pressure rises, thereby raising the mean arterial pressure (MAP); meanwhile, the cerebral autoregulation range keeps the cerebral blood flow constant. Metabolic regulation is affected by the increase in brain glucose utilization and lactate levels. The glucose metabolism is systemically increased in skeletal muscle but not significantly increased in the brain, only in areas where increased brain activity due to exercise is detected [2]. Although neurogenic regulation of cerebral blood vessels is possible through sympathetic innervation, it is not well understood whether this phenomenon plays an independent regulatory role in the control of brain flow during exercise. Although autoregulation, metabolic, and neurogenic regulation appear to be three distinct mechanisms, cerebral circulation is maintained through a complex system of these and other minor systemic processes [37]. Our results provide insights into how, in addition to the regulatory processes described by numerous functional studies, histological changes protect the complex system of cerebral circulation.

### 4.1. Exercise Differences

The function and structure of cerebral blood flow and cerebral blood vessels have been investigated by several groups [3,4]. An important question is about the effect of aging on the pial and intracerebral blood vessels and whether these processes can be influenced by exercise. Rzechorzek et al. investigated the effects of aerobic exercise on the pial collateral reticular formation. Aging causes a reduction in the lumen and number of the pial collaterals, enhancing network resistance and thus the risk of stroke and increasing the extent of tissue damage caused by stroke through damage to the collateral network. Pial collaterals show faster aging than other vessels of a similar size, and aging results in a decrease in endothelial nitric oxide synthase (eNOS) activity and a decrease in smooth muscle density in the vessel wall. They found that exercise training did not cause a narrowing of the lumens of the pial collaterals, nor did it decrease their number or increase their flow after acute infarction. They also observed an increase in the activity of eNOS and antioxidant enzymes [38]. Zhang et al. investigated circulatory changes in stroke as a result of exercise in their animal model. They observed that exercise created a denser microcirculatory network in the stroke area through the amplification of angiogenic factors. This resulted in an increased blood flow in the ischemic area and decreased its overall volume. Through the increase in blood flow, functional improvement also showed a better prognosis compared to the non-athletic animals [39]. It is important to highlight that it is not only in functional animal models that we can see the positive effects of exercise on cerebral circulation and cognitive and motor function improvement post-stroke. In a recently published meta-analysis by Mou et al., the question is not whether exercise helps in the rehabilitation of stroke patients, but what type of exercise program can achieve better results [40]. Altered vessel wall elasticity has been shown to be a major factor in the aging process. One theory describing the development of dementias, the vascular hypothesis, identifies vascular damage to cerebral vessels as the pathological cause of dementia, with damage to the neurovascular unit (NVU) leading to local circulatory disturbances [41,42,43,44,45]. Aging reduces the elasticity of both peripheral and large cerebral arteries, altering local flow conditions, increasing oxidative stress, and changing shear forces. These altered conditions can lead to endothelial damage and reduced vascular reactivity in both large cerebral vessels and in intracerebral small arterioles [11]. Intensive exercise completed 4–5 times a week has been found to prevent this aging-associated arterial stiffening, with regular intense aerobic exercise possibly even reversing it. Light aerobic activity alone was found to be insufficient [12,13,14,15,16].

Our present experiment confirms the lesions observed in functional studies with histological evidence. In both sexes, we observed an increase in the density of elastic fibers in both intracerebral and pial arterioles in the exercising animals compared to the sedentary groups. It can be assumed that the increased eNOS activity in these animals also resulted from exercise, leading to these beneficial histological changes. However, we did not find a significant increase in the density of smooth muscles in response to exercise training. These results show that exercise training increases the elasticity of arterioles, which, based on the functional studies discussed earlier, is thought to have a protective effect against vascular aging. However, by interpreting these results, we must take into account that our experiments were performed on young healthy animals.

### 4.2. Sex Differences

There is considerable evidence that the female sex is more protected against major cardiovascular events compared to men, predominantly in the case of premenopausal women [17,18,19]. There are many studies investigating the difference in vascular function between the sexes, but these are mainly based on studies of peripheral blood vessels rather than cerebral blood vessels [20]. Pak et al. have investigated the difference in vascular reactivity between the sexes in rat tail arteries. Their study revealed that different factors influence reactivity (eNOS activity, K^+^, and prostacyclin-dependent relaxation [21]) in females under mechanical stress, or acetylcholine influences vascular tone by both NOS and K^+^-regulated mechanisms. In contrast, in males, only NOS-induced vasodilation was significant [22]. Zhang et al. have shown in diabetic rats that there are also sex differences in the development of diabetic angiopathy. In females, K^+^-dependent relaxation becomes dominant in response to the vasodilatory effects of diabetes, whereas in males, this is the dominant process by default [23]. Only a few groups have addressed the sex differences affecting the reactivity of cerebral blood vessels. Chrissobolis et al., studying the basilar artery, observed that K^+^-mediated vasodilation was stronger in female animals than in males, an effect attributed to the presence of higher estrogen concentrations [24]. Arrick et al. studied the reactivity of brain arterioles during exercise. Their results showed that NOS-dependent brain arteriolar reactivity was greater in female animals in both the exercising and sedentary groups compared with male animals. It was hypothesized that this nonspecific difference may be due to the intensity of nitric oxide synthesis, and the effects of other vasodilatory processes present in females. In addition, it has been described that the reactivity of the pial arterioles was not increased in either sex in exercising animals [20].

In an earlier work from our laboratory, increased coronary artery elastica density has been proven in both sexes [29], while in renal arteries, exercise decreased smooth muscle actin and elastic fiber density in females compared to males [30]. The difference obviously is the result of the different roles of these vessels during heavy physical activity in which cerebral flow is maintained while renal circulation should be reduced. In gracilis muscle arterioles, being musculocutaneous arteries directly involved in exercise, training reduced the elastic modulus, the myogenic tone, and smooth muscle density in males, while it increased distensibility, myogenic tone, and smooth muscle density in females [28]. Our findings on cerebral blood vessels revealed similar sex-specific morphological differences. These sexual differences can be either the effects of sex hormones, or they can be results of the fact that similar training means different levels of workload for both sexes.

Our present results show that sex difference determines the response to exercise, but it already shows differences in the histology of the vascular wall in sedentary animals.

Comparing the two sex sedentary groups, we found a significant difference in the smooth muscle density of pial arterioles. The male sedentary animals had significantly higher smooth muscle density in the pial arterioles. However, this difference disappeared with exercise. An interesting difference is that when comparing the training groups, we observed an increase in smooth muscle density in the intracerebral vessels in male animals. It can be assumed that the female animals were more affected by the training, and, therefore, the difference seen in the pial vessels could have been compensated. In addition, we observed an increase in elastic fiber density in the pial vessels as a result of exercise compared to the male exercising animals. The increase in vascular elasticity is more pronounced overall in female animals.

## 5. Strengths and Limitations

In interpreting the results, we must take into account that we cannot associate our histological or molecular analysis results with our own functional test results. In addition, the experiment was performed in healthy rats and cannot be fully extrapolated to the human vascular aging process. It does, however, provide a good insight into the basis of a complex process and how exercise causes histological changes in brain arterioles.

In the present experiment, we did not use the perfusion fixation technique commonly used in vascular histology studies but instead used the immersion tissue fixation technique. The vasoconstrictive effect of this technique was taken into account when calculating the true diameters of the vessels.

We would also like to highlight that our research aims to fill this gap that functional studies do not answer.

## 6. Conclusions

This study is the first attempt to characterize the histological changes in cerebral blood vessels during chronic exercise, which could be suspected based on functional studies. Our findings, compared with the existing literature, suggest that exercise-induced histological changes in cerebral blood vessels, starting at the level of the smallest arterioles, play an important role in the regulation of local circulation. Exercise-induced morphological changes, such as an increase in elastic fiber density, benefit the physiological regulating function of both intracerebral and pial vessels. Notably, exercise induced more significant changes in female animals, highlighting the influence of sex differences on exercise’s effects on the body. In conclusion, exercise induces changes in the expression of contractile and passive force-bearing histological elements of cerebral resistance arteries that improve the physiological function of cerebral vessels and help maintain cerebral blood flow. The substantial difference we found between male and female animals points out the significance of sexual hormones in the process of vascular adaptation to heavy training, being present even in the cerebral resistance vessels.

## Figures and Tables

**Figure 1 life-15-01304-f001:**
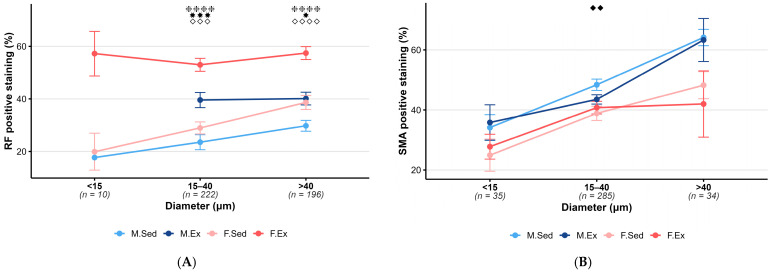
**Exercise and sex effect on pial arterioles.** (**A**) Mean RF-positive staining by diameter and group for pial arterioles (mean ± SEM). Training increased the elastic fiber density in both sexes, and trained females had higher elastic fiber density than trained males. Sample sizes (arterioles) are shown below each diameter. Statistical significance was assessed by ANOVA with Tukey’s post hoc test. Symbols indicate pairwise comparisons: ✸ *p* < 0.05 (M.Sed vs. M.Ex), ✸✸✸ *p* < 0.001 (M.Sed vs. M.Ex), ❈❈❈❈ *p* < 0.0001 (F.Sed vs. F.Ex), ◇◇◇ *p* < 0.001 (M.Ex vs. F.Ex), and ◇◇◇◇ *p* < 0.0001 (M.Ex vs. F.Ex). Post hoc power analysis (α = 0.05) by diameter category showed the following: f = 1.01 (62% power, <15 µm), f = 0.57 (100% power, 15–40 µm), and f = 0.56 (89% power, >40 µm). (**B**) Mean SMA-positive staining by diameter and group for pial arterioles (mean ± SEM). Smooth muscle density was significantly higher in male control animals than in female controls in the 15–40 um range. Sample sizes (arterioles) are shown below each diameter; statistical significance was assessed by ANOVA with Tukey’s post hoc test. Symbols indicate pairwise comparisons: ◆◆ *p* < 0.01 (M.Sed vs. F.Sed). Post hoc power analysis (α = 0.05) by diameter category showed the following: f = 0.33 (28% power, <15 µm), f = 0.21 (85% power, 15–40 µm), and f = 0.55 (72% power, >40 µm). **Abbreviations**: Male sedentary (M.Sed), male exercising (M.Ex), female sedentary (F.Sed), and female exercising (F.Ex).

**Figure 2 life-15-01304-f002:**
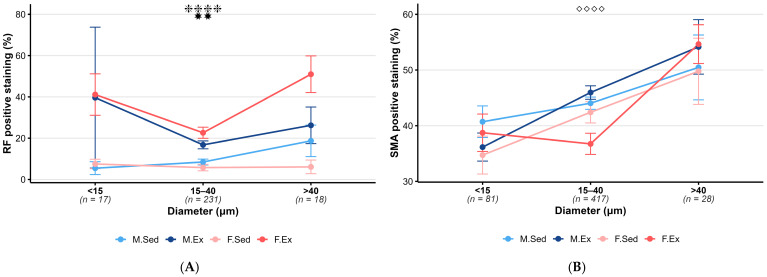
**Exercise and sex effect on intracerebral arterioles.** (**A**) Mean RF-positive staining by diameter and group for intracerebral arterioles (mean ± SEM). Both sexes showed an increase in elastic fiber density after exercise. Sample sizes (arterioles) are shown below each diameter. Statistical significance was assessed by Kruskal–Wallis with Dunn post hoc test. Symbols indicate pairwise comparisons: ✸✸ *p* < 0.01 (M.Sed vs. M.Ex), ❈❈❈❈ *p* < 0.0001 (F.Sed vs. F.Ex). Post hoc power analysis (α = 0.05) by diameter category showed the following: f = 0.93 (80% power, <15 µm), f = 0.43 (80% power, 15–40 µm), and f = 0.87 (78% power, >40 µm), respectively. (**B**) Mean SMA-positive staining by diameter and group for intracerebral arterioles (mean ± SEM). Smooth muscle density is greater in trained males compared to trained females in the intracerebral arteries. Sample sizes (arterioles) are shown below each diameter. Statistical significance was assessed by ANOVA with Tukey’s post hoc test. Symbols indicate pairwise comparisons: ◇◇◇◇ *p* < 0.0001 (M.Ex vs. F.Ex). Post hoc power analysis (α = 0.05) by diameter category showed the following: f = 0.13 (13% power, <15 µm), f = 0.22 (98% power, 15–40 µm), and f = 0.16 (8% power, >40 µm), respectively. **Abbreviations:** Male sedentary (M.Sed), male exercising (M.Ex), female sedentary (F.Sed), and female exercising (F.Ex).

## Data Availability

The raw data supporting the conclusions of this article will be made available by the authors on request.

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
