# Peer review of "Cerebral Resistance Artery Histological Remodeling After Training—Sex Differences"

_life, 2025, doi:10.3390/life15081304_

Round 1

Reviewer 1 Report

Comments and Suggestions for Authors

Dear authors,

the manuscript is interesting. I have one question:

Line 305-307 The authors offer two versions. Is there any evidence that similar training means different level of workload for both sex in the rats?

Author Response

Dear Reviewer1,

Thank you for your thorough revision of our manuscript.

We greatly appreciate both your positive comments and your helpful suggestions.

Our replies are as follows:

Is there any evidence that similar training means different level of workload for both sex in the rats?

Several groups have been used to study the effects of regular exercise in animal models. When detecting sex differences, it is important that the training programme is the same in both sexes. The adaptation of the myocardium to increased exercise-induced workload is a well-researched area, and we would like to use this example to answer this question. In several studies, including the experiments of our group, it has been described that male and female rats undergoing the same exercise programme had significantly greater myocardial hypertrophy in females compared to male exercise-trained animals. These data suggest that exercise-induced stress may be sex-differentiated and that this is not only reflected in the heart but also in adaptation mechanisms of other organ systems.

References

  1. Konhilas, J.P.; Maass, A.H.; Luckey, S.W.; Stauffer, B.L.; Olson, E.N.; Leinwand, L.A. Sex modifies exercise and cardiac adaptation in mice. Am J Physiol Heart Circ Physiol 2004, 287, H2768-2776, doi:10.1152/ajpheart.00292.2004.
  2. Foryst-Ludwig, A.; Kreissl, M.C.; Sprang, C.; Thalke, B.; Böhm, C.; Benz, V.; Gürgen, D.; Dragun, D.; Schubert, C.; Mai, K.; et al. Sex differences in physiological cardiac hypertrophy are associated with exercise-mediated changes in energy substrate availability. Am J Physiol Heart Circ Physiol 2011, 301, H115-122, doi:10.1152/ajpheart.01222.2010.
  3. Török, M.; Monori-Kiss, A.; Pál, É.; Horváth, E.; Jósvai, A.; Merkely, P.; Barta, B.A.; Mátyás, C.; Oláh, A.; Radovits, T.; et al. Long-term exercise results in morphological and biomechanical changes in coronary resistance arterioles in male and female rats. Biol Sex Differ 2020, 11, 7, doi:10.1186/s13293-020-0284-0.

We would like to thank our Reviewer for the careful and detailed overview and useful advices.

We hope that the revised manuscript will be acceptable for publication in Your highly esteemed Journal.

Kind regards,

Marianna Török

Reviewer 2 Report

Comments and Suggestions for Authors

Hainzl T et al investigated the effect of exercise on cerebral resistance artery morphology, and differences between males and females in rats. There are some major concerns:

  • The study uses a very small number of animals (n = 5–6 per group), which limits statistical power and raises concerns about the reliability of the results and reproducibility. The analysis includes multiple comparisons and subgroup analyses by sex and vessel size, but there's no power analysis provided to justify that the study is adequately powered to detect the reported differences.
  • While the introduction touches on the general benefits of exercise and vascular remodeling, the specific hypothesis and rationale behind focusing on sex differences in cerebral resistance arteries are not clearly stated. Why is this research gap important, and how does the study address a novel question beyond what's already known?
  • The image processing pipeline and thresholds (for DAB and RF positivity) appear arbitrarily chosen or based on unpublished “preliminary data.” There is no validation of this method against a gold standard, nor any discussion of inter- or intra-observer reliability. Please provide justification for thresholds, include validation metrics, and describe whether the analysis was blinded.
  • The claim that “exercise may help prevent vascular aging and preserve cognitive function” is overstated, given that only histological data in healthy young rats are presented, with no cognitive or functional measures. Rephrase conclusions to match the scope of the data and emphasize the limitations of translational relevance.
  • While the figures are informative, the legends are dense and hard to interpret, and the sample sizes per vessel type are somewhat buried in the text. Also, error bars and statistical annotations should be clarified. Please improve figure clarity and ensure all results shown correspond to described statistical tests. Include p-values where applicable.

Author Response

Dear Reviewer2,

Thank you for your thorough revision of our manuscript.

We greatly appreciate both your positive comments and your helpful suggestions.

Our replies are as follows:

The study uses a very small number of animals (n = 5–6 per group), which limits statistical power and raises concerns about the reliability of the results and reproducibility. The analysis includes multiple comparisons and subgroup analyses by sex and vessel size, but there's no power analysis provided to justify that the study is adequately powered to detect the reported differences.

We calculated the statistical power of our calculations and stated them in the results and in the figure legends.

While the introduction touches on the general benefits of exercise and vascular remodelling, the specific hypothesis and rationale behind focusing on sex differences in cerebral resistance arteries are not clearly stated. Why is this research gap important, and how does the study address a novel question beyond what's already known?

Thank you for your suggestion, we made the necessary changes in the introduction between line 81-90.

The image processing pipeline and thresholds (for DAB and RF positivity) appear arbitrarily chosen or based on unpublished “preliminary data.” There is no validation of this method against a gold standard, nor any discussion of inter- or intra-observer reliability. Please provide justification for thresholds, include validation metrics, and describe whether the analysis was blinded.

Staining technique was standardized, using an automated device. Scanning was performed after stabilization of light intensity , for all sections in a relative short time. Thickness of sections was chosen at 5 micrometers, which is not optimal for sharpness of structures, but ensures more stain in the light pathway. As both smooth muscle and elastic structures can be thinner than 5 micrometers, more or less of these components can be present in the column of a pixel. Calibration was made as follows: Unanimously elastic and DAB positive territories, respectively, were cut from stochastically chosen sections (several millions of pixels). For comparisons unanimously not elastic (or DAB negative) territories have also been cut from the pictures. Red, green and blue intensities on the 1-255 color intensity scale have been red for each pixel and two-dimensional color intensity histograms constructed. These diagrams demonstrated that for (our standard staining procedure) SMA DAB R/B>1.2 pixels give a good, objective definition for DAB positivity. In case of RF, the purple color depresses the green component, G<40 can be defined as containing mostly elastin component. Such objective criteria have been used throughout the evaluation, excluding any further subjectivity.

The selection of the arterial cross sections and the colorimetric analysis was double blinded.

References:

  1. Hadjadj, L.; Monori-Kiss, A.; Horváth, E.M.; Heinzlmann, A.; Magyar, A.; Sziva, R.E.; Miklós, Z.; Pál, É.; Gál, J.; Szabó, I.; et al. Geometric, elastic and contractile-relaxation changes in coronary arterioles induced by Vitamin D deficiency in normal and hyperandrogenic female rats. Microvasc Res 2019, 122, 78-84, doi:10.1016/j.mvr.2018.11.011.
  2. Hetthéssy, J.R.; Tőkés, A.M.; Kérész, S.; Balla, P.; Dörnyei, G.; Monos, E.; Nádasy, G.L. High pressure-low flow remodeling of the rat saphenous vein wall. Phlebology 2018, 33, 128-137, doi:10.1177/0268355516688984.

The claim that “exercise may help prevent vascular aging and preserve cognitive function” is overstated, given that only histological data in healthy young rats are presented, with no cognitive or functional measures. Rephrase conclusions to match the scope of the data and emphasize the limitations of translational relevance.

Thank you for the suggestion we rephrased it and added a Strengths and Limitations paragraph to our manuscript.

While the figures are informative, the legends are dense and hard to interpret, and the sample sizes per vessel type are somewhat buried in the text. Also, error bars and statistical annotations should be clarified. Please improve figure clarity and ensure all results shown correspond to described statistical tests. Include p-values where applicable.

Thank you for the suggestion we corrected it.

We would like to thank our Reviewer for the careful and detailed overview and useful advices.

We hope that the revised manuscript will be acceptable for publication in Your highly esteemed Journal.

Kind regards,

Marianna Török

Reviewer 3 Report

Comments and Suggestions for Authors

1. Abstract

  • The abstract is informative but could be improved by including exact p-values or effect sizes to emphasize the statistical strength of findings.

2. Introduction

  • While comprehensive, the introduction could benefit from a more concise structure, especially in summarizing the known literature on sex-specific cerebrovascular adaptations.

3. Discussion Depth

  • The discussion covers key findings but occasionally repeats earlier results. Instead, the authors could expand on potential molecular mechanisms (e.g., estrogen-related pathways, NO signaling).

  • The clinical translation of findings (e.g., implications for stroke prevention or cognitive aging) should be discussed more thoroughly.

4. Terminology Consistency

  • The manuscript switches between “control” and “sedentary” terminology. Standardizing this language would improve clarity.

  • Consider harmonizing vessel size references (e.g., "diameter > 15 µm") throughout for easier comparison.

5. Limitations Section

  • The paper lacks a dedicated limitations paragraph. It would be helpful to acknowledge:

  • The small sample size per subgroup.

  • Limitations in extrapolating rat data to human cerebrovascular physiology.

  • Absence of molecular analysis to complement histological findings.

Author Response

Dear Reviewer3,

Thank you for your thorough revision of our manuscript.

We greatly appreciate both your positive comments and your helpful suggestions.

Our replies are as follows:

  1. Abstract

The abstract is informative but could be improved by including exact p-values or effect sizes to emphasize the statistical strength of findings.

Thank you for your suggestion, we corrected it.

  1. Introduction

While comprehensive, the introduction could benefit from a more concise structure, especially in summarizing the known literature on sex-specific cerebrovascular adaptations.

Thank you for your suggestion, we corrected it. Line 71-80.

  1. Discussion Depth

The discussion covers key findings but occasionally repeats earlier results. Instead, the authors could expand on potential molecular mechanisms (e.g., estrogen-related pathways, NO signaling).

The clinical translation of findings (e.g., implications for stroke prevention or cognitive aging) should be discussed more thoroughly.

Thank you for your suggestion, we corrected it.

  1. Terminology Consistency

The manuscript switches between “control” and “sedentary” terminology. Standardizing this language would improve clarity.

Consider harmonizing vessel size references (e.g., "diameter > 15 µm") throughout for easier comparison.

Thank you for your suggestion, we corrected it.

  1. Limitations Section

The paper lacks a dedicated limitations paragraph. It would be helpful to acknowledge:

The small sample size per subgroup.

Limitations in extrapolating rat data to human cerebrovascular physiology.

Absence of molecular analysis to complement histological findings.

Thank you for your suggestion, we dedicated a paragraph to the limitations of our study befor the Conclusions.

We would like to thank our Reviewer for the careful and detailed overview and useful advices.

We hope that the revised manuscript will be acceptable for publication in Your highly esteemed Journal.

Kind regards,

Marianna Török

Round 2

Reviewer 2 Report

Comments and Suggestions for Authors

Authors have answered all comments point by point.

Author Response

Thank you for your review.